# Mechanical Properties and Wear Susceptibility Determined by Nanoindentation Technique of Ti13Nb13Zr Titanium Alloy after “Direct Laser Writing”

**DOI:** 10.3390/ma16134834

**Published:** 2023-07-05

**Authors:** Magdalena Jażdżewska, Beata Majkowska-Marzec, Andrzej Zieliński, Roman Ostrowski, Aleksandra Frączek, Gabriela Karwowska, Jean-Marc Olive

**Affiliations:** 1Department of Biomaterials Technology, Faculty of Mechanical Engineering and Ship Technology, Institute of Manufacturing and Materials Technology, Gdańsk University of Technology, 80-233 Gdańsk, Poland; magdalena.jazdzewska@pg.edu.pl (M.J.); gabriela.karwowska1@gmail.com (G.K.); 2Institute of Optoelectronics, Military University of Technology, 00-908 Warszawa, Poland; roman.ostrowski@wat.edu.pl; 3CNRS, Institute of Mechanics and Engineering, University of Bordeaux, 33400 Talence, France

**Keywords:** laser surface treatment, titanium alloys, wear resistance, roughness, mechanical properties

## Abstract

Laser treatment has often been applied to rebuild the surface layer of titanium and its alloys destined for long-term implants. Such treatment has always been associated with forming melted and re-solidified thin surface layers. The process parameters of such laser treatment can be different, including the patterning of a surface by so-called direct writing. In this research, pulse laser treatment was performed on the Ti13Nb13Zr alloy surface, with the distance between adjacent laser paths ranging between 20 and 50 µm. The obtained periodic structures were tested to examine the effects of the scan distance on the microstructure using SEM, the roughness and chemical and phase composition using EDS and XRD, and the mechanical properties using the nanoindentation technique. After direct laser writing, the thickness of the melted layers was between 547 and 123 µm, and the surface roughness varied between 1.74 and 0.69 µm. An increase in hardness was observed after laser treatment. The highest hardness, 5.44 GPa, was obtained for the sample modified with a laser beam spacing of 50 µm. The value of the distance has been shown to be important for several properties and related to a complex microstructure of the thin surface layer close to and far from the laser path.

## 1. Introduction

Titanium and its alloys are increasingly used in different branches of the economy. One of the most important applications is the fabrication of long-term implants, mainly dental and joint ones. However, despite the high strength-to-density ratio, low Young’s modulus for β alloys, and remarkable biocompatibility, they also have some weaknesses, such as relatively low hardness and wear resistance, and also bioactivity. To improve this behavior, various surface modifications have been developed to remodel the surface, introduce different elements to create interstitial solutions, as well as interstitial and intermetallic phases, and deposit coatings [1,2,3,4,5,6]. Among them, the most important solutions that focus on the improvement of the mechanical characteristics of the surface include different forms of laser treatments, resulting in enhanced mechanical characteristics, wettability, and biological properties [7,8,9,10,11,12].

Laser treatment can be carried out using various techniques. Generally speaking, these techniques can be divided into three groups: direct laser interference patterning (DLIP), laser-induced periodic surface structures (LIPSS) [7,8,13], and direct laser writing (DLW). Each of them has its advantages and disadvantages. With DLIP and LIPSS techniques, structures with very small periods can be produced on the surface, and the limitation is the wavelength of laser radiation. In the case of the DLW technique, the period of the structure is limited by the size of the laser spot in the focus and the resolution of the scanning system. Therefore, the periods of the obtained surface structures are usually longer. In the LIPSS and DLIP methods, the size of the surface on which the periodic distribution is created is limited by the size of the laser spot and the available energy in the laser pulse. Therefore, laser modifications are limited to relatively small areas, on the order of a square centimeter. In addition, the DLIP technique requires beams with very good quality and high coherence. In contrast, in the DLW technique, where the only limitation is the working area of the scanning system, very large surfaces can be subjected to laser modification. The scanning required here, however, causes the production time of the periodic structure to be much longer compared to the DLIP technique, where periodic patterns are created by irradiating with only a few pulses. The LIPSS method, in turn, requires exposures consisting of even several thousand laser pulses. Its main advantage, however, is simplicity, because the experimental setup basically consists of a suitable laser and possibly a single focusing lens. Laser systems used in the DLIP method are usually complex and require time-consuming adjustment, and due to the interference processes occurring here, mechanical stability (elimination of vibrations of the optical table) and the stability of environmental conditions (air temperature and humidity) become very important. System adjustment processes in the DLW method are much easier, and the systems themselves are highly resistant to changes in environmental conditions in a relatively wide range. In conclusion, the production speed and minimum characteristic size (period) obtained by DLIP and LIPSS methods have resulted in the fact that most of the work on laser surface modification for biomedical engineering has focused on these two techniques. However, when a large modified area is needed, and the characteristic size is not so critical, it is worth using the simpler DLW method. Therefore, the current work focuses on surface modification using the DLW method and what changes it causes in the mechanical properties of titanium alloys.

Laser treatment causes the significant remodeling of the surface topography. Such structural modification results in the appearance of higher roughness and many surface imperfections. The latter include micro-grooves and micro-protrusions [14], nanometric-size cavities [15], and even cracks [16]. The chemical composition is slightly sensitive and can manifest in a local oxygen increase within the surface layer [17,18], specifically in micro-protrusions rather than in micro-grooves [14]. In addition, excessive oxygen promotes the transformation of Ti and Ti_2_O_3_ into TiO_2_ rutile/anatase dioxide [17].

Laser-enhanced modification distinctly changes the surface properties, such as hardness, wear resistance, and resistance to plastic deformation, determined in nanoindentation tests [18]. Following laser treatment, the appearance of surface residual stresses, either compressive or tensile, has been observed [17,19,20,21].

In the case of pulsed lasers, the most important process parameters are the energy density (fluence), average power, pulse repetition rate, and scanning speed [19]. The roughness increased at a lower laser speed and higher frequency [7]. Hardness was shown to increase [22,23] or was not significantly influenced [24] by laser treatment. Increasing tensile strength was reported [23,24], but Young’s modulus remained stable [24] or increased [23]. The mechanical properties and wear resistance were not related to each other when determined in the nanoindentation test [25]. All these phenomena were attributed to the occurrence of melting, the rapid crystallization of thin subsurface layers, and even cavitation [23]. The effect of laser treatment on corrosion resistance has seldom been investigated and is reported to be positive [26].

Laser treatment positively influenced the water contact angle values, a physical indicator of biocompatibility. Hydrophilicity, desired for the adhesion of osteoblasts on the surface, or even superhydrophilicity was observed, particularly when using a femtosecond laser [14,16,17,27]. These effects were explained by the appearance of several microcolumns, nanopillars, and ripples on the surface following laser irradiation [7,8]. Bioactivity, determined directly by the deposition of an apatite layer in SBF (simulated body fluid), was observed after Yb fiber laser treatment [28].

Among biological properties, cell proliferation and differentiation were improved [23], and the absorption of proteins [29] was shown. Osseointegration was positively influenced by laser irradiation, as assessed by in vivo experiments on the strength of the implant’s fixation in bone [18] and the bone–implant contact area [23]. No advantages of laser treatment compared to nitriding were, however, noticed [15]. Cell activity was reduced, and cell proliferation was retarded on a superhydrophilic surface [17]. A bacteria-killing effect was also observed owing to the impairment of the adhesion of numerous bacteria after LIPSS treatment with a femtosecond laser and, simultaneously, the improved adhesion of gingival tissue [30]. DLIP can be useful for separately affecting the attachment of osteoblasts and bacteria [7,8].

There have been a variety of different laser-associated treatments focused on increasing several mechanical properties, primarily hardness and wear properties. This research is a novel approach in this field, as it characterizes the technique of direct laser writing for titanium and other metals, which has seldom been studied. This technique is promising, as it presumably does not cause much damage to the surface layer. The aim of this research is to verify whether DLW can have a significant effect on the mechanical behavior of a Ti alloy with the limited rebuilding of the surface layer. The performed surface modification was made at different distances between the laser beam scans, and the mechanical behavior was characterized and analyzed based on changes in the microstructure. The present investigation proves that this form of laser treatment can effectively and positively affect surface properties.

## 2. Materials and Methods

### 2.1. Preparation of Samples

Ti13Nb13Zr titanium alloy (Table 1) samples (Xi’an SAITE Metal Materials Development Co., Ltd., Xi’an, China) were prepared for testing. The samples were cut from a rod with a diameter of 28 mm and delivered annealed. Then, they were sanded with sandpaper on the SAPHIR 330 sander and polisher (ATM; delivered by Prospecta, Warszawa, Poland). The treatment was performed using sandpaper with a decreasing gradation of the grain from 180 to 2000. The final stage was polishing with a diamond paste with 3 μm granulation.

### 2.2. Laser Treatment

The modification of the samples was carried out by means of the direct laser writing method. The experimental setup consisted of an ytterbium fiber laser coupled with a galvanometer scanner with an F-theta lens with a focal length of 160 mm. The scanning speed was 5 mm/s, and the scan spacing was changed from 20 μm to 50 μm. The pulse energy was constant and equal to 530 mJ. To achieve a good-quality line pattern, the repetition rate was set to 1 kHz. Four different laser beam spacings were used during processing: 20 (LT 20), 30 (LT 30), 40 (LT 40), and 50 (LT 50) μm. To reduce surface oxidation during laser processing, argon 5.0 was used (Linde Gaz Poland Ltd., Krakow, Poland).

### 2.3. Topography, Chemical and Phase Composition, and Cross-Section Analyses

The SEM microscope was used to study the surface structure of the laser-treated Ti13Zr13Nb alloy (JEOL JSM-7800 F, JEOL Ltd., Tokyo, Japan). The chemical composition on the surface of the samples was analyzed using the X-ray energy dispersion (EDS) spectrometer (Octene Elite 25, EDAX, Mahwah, NJ, USA) attached to the SEM microscope. An analysis of the chemical composition was performed for an area of 400 × 300 μm.

The thickness of the remelted surface layers was assessed on cross-sections of samples that were ground, polished, and etched with Kroll’s reagent (0.06 mm^3^ HF, 0.12 mm^3^ HNO_3_, 50 mm^3^ distilled water). An optical microscope was used for examination (UC50, Olympus Europa SE&Co. KG, Hamburg, Germany).

Phase composition analysis was performed using a PHILIPS X’PERT-PRO (PHILIPS, Almelo, The Netherlands) diffractometer with a copper lamp. The X-ray method using Cu Kα radiation was performed to analyze the phase composition, with a wavelength of radiation λKα1 = 0.15406 nm and λKα2 = 0.15444 nm.

### 2.4. Roughness Examinations

The surface roughness profile was measured using the stylus method, where a moving needle recorded changes in surface height at a consistent velocity. The measurement was carried out using the Hommel Etamic Waveline profilograph (JENOPTIK, Dresden, Germany). The measurement section had a length of 3.75 mm. Additionally, each roughness parameter (Ra) result represents the arithmetic average of three measurements.

### 2.5. Mechanical Tests

Microhardness measurements on cross-sections in the near-surface zone of the melted layer were made using the Vickers method with a microhardness tester (FM-800 Future-Tech., Kawasaki, Japan). The samples were subjected to a load of 10 gf for 10 s-HV 0.01. For each of the samples, measurements of the microhardness of the near-surface zone (LT) and the base material (BM) were made.

The nanoindentation study was performed using a NanoTest Vantage (MicroMaterials, Wrexham, UK). A Berkovich diamond indenter with a point angle of 142.4° was used. For the samples after laser treatment (LT) and the comparative sample (BM), 6 measurements were made. The loading and unloading time for each measurement was 20 s. After reaching the maximum force (200 mN), the sample was held for 5 s to stabilize the maximum indentation depth. In each of the measurement sites, a cavity was made with 10 different force values (10, 20, 40, 60, 80, 100, 125, 150, 175, and 200 mN).

Based on the values measured during the study and the values available in the literature, the values of Young’s modulus were calculated according to Formula (1) (Oliver–Pharr method).
(1)$$E_{s}=\frac{(1-v_{s}^{2})\cdot E_{r}\cdot E_{i}}{{(E}_{i}-E_{r}(1-v_{i}^{2}))}$$
where:

E_r_—reduced Young’s modulus (measured value);

E_s_—Young’s modulus of the tested material (search value);

E_i_—Young’s modulus of the indenter material (1141 GPa, manufacturer’s data);

v_s_—Poisson’s ratio of the tested material (0.36) [31];

v_i_—Poisson’s ratio of the indenter material (0.07, manufacturer’s data).

To evaluate the tribological resistance based on the results of the nanoindentation test, the following parameters are defined:The wear resistance factor is calculated using the H/E_r_ formula and determines the material’s resistance to elastic deformation.The ratio determining the material’s resistance to plastic deformation is expressed by the formula H^3^/E_r_^2^. It determines the ability of a material to dissipate energy after deformation under a load [32,33].In the formulas, H is hardness, and E_r_ is Young’s reduced modulus.

## 3. Results and Discussion

### 3.1. Surface Morphology, Topography, and Chemical and Phase Composition

Figure 1 illustrates the topography of samples subjected to direct laser writing (DLW). The only change on each surface is porosity. Such imperfections as cracks, previously observed after laser irradiation [34,35], do not appear. This may be evidence of the presence of compressive stresses within the thin subsurface layer [36,37] and the absence of a brittle zone in which tensile residual stresses might occur [20]. DLW laser treatment seems to be more subtle and does not introduce significant damage to the surface layer, which is an advantage of this method compared to some other forms of laser irradiation and pulse laser power values such as 1000 W [21].

Table 2 shows the chemical composition of the samples. In the base material, the content of zirconium increased, but the differences are minor depending on the treatment parameter, and their content is above 14% by weight. An increase in niobium can be observed for the 50 µm setting, where it is above 16%. The results are the average value of the tested variable. The relative increase in the contents of Zr and Nb at the expense of Ti can be easily explained by titanium having the highest vaporization pressure among the three alloying elements, and its loss during laser treatment is certainly the highest [38].

Figure 2 presents the phase and chemical composition of base and remelted materials. An XRD analysis of the base material (BM) confirmed the occurrence of only Tiα (JCPDS no. 44-1294) and Tiβ (JCPDS no. 44-1288) phases. For each remelted specimen (LT), the main phases that appear are TiαTiβ, TiO_2_, and NbO_2_. The presence of titanium oxide (TiO_2_) is proved by the peaks at 36.98° and 53.99° angles (JCPDS no. 21-1272), with NbO_2_ appearing at 62.36° (JCPDS no. 44-1053) and Nb_2_O_5_ observed at a 70.69° angle (according to JCPDS no. 30-0873).

### 3.2. Thickness of the Surface Layers

Figure 3 shows the microstructures of the cross-sections of the samples. For each of the samples, six measurements of the thickness of the melted layer were made. The average values, along with their standard deviations, are presented in Table 3.

The cross-sections show a zonal structure. A significant difference is visible between the near-surface zone and the other zones. It is characterized by the fragmentation of the needle structure at the very surface. The middle zone is characterized by a fragmented, coniferous structure. The needles are larger and thicker than in the near-surface zone. The transition zone is characterized by longer and thinner needles.

The thicknesses of the surface layers were as follows: for sample LT 20—547.3 μm; LT 30—453.9 μm; LT 40—405.55 μm; and LT 50—123.9 μm. These values are lower than those usually reported, such as a 900 µm zone at 500 W pulse laser power [39]; above 500 µm using the Nd:YAG laser [24]; or comparably, 400–500 µm at 500 W [36]. Based on observed cross-sections, it can be said that the thickness of laser-melted surface layers and particular zones in the whole layer decreased with the increasing distance of subsequent laser beam passages, an obvious effect of laser energy locally decreasing over and beneath the surface.

### 3.3. Microhardness of Cross-Sections of the Surface Layer after Laser Treatment

The results of microhardness measurements on cross-sections in the near-surface zone of the remelted layer are presented in Table 4. The microhardness of the Ti13Nb13Zr BM titanium alloy on the Vickers scale was 254 HV. As a result of laser treatment, the samples were hardened. The average hardness of the near-surface zone was 3.034 GPa for the LT 20 sample, 0.3632 GPa for LT 30, 0.372 GPa for LT 40, and 0.3867 GPa for LT 50. The difference in the hardness of the near-surface zone between the LT 30 and LT 40 samples was insignificant. Based on the results presented in Table 4, it can be concluded that the hardness of the near-surface zone increases with the increase in the laser beam spacing. The lower-energy flow makes the zone-affected zone depth smaller, and higher energy makes it larger. Therefore, the highest hardness appears at some distance from the surface; e.g., Katahira et al. [28] observed a surface hardness of 1.618 GPa, which increased to 2.511 GPa at 3 μm and further decreased to 9 μm. It is then assumed that the hardness peak is visible at some depth, likely increasing with lowering energy. However, this is a speculative assumption, and the effects of the possible dissolution and precipitation of intermetallic phases related to the power and temperature or of thermal stresses should be taken into account. This currently unexplained observation will be studied in the future.

### 3.4. Roughness

Table 5 lists the roughness values shown as arithmetic means and calculated standard deviations. Roughness is significantly dependent on the distance between laser passages. A decrease in roughness with increasing distance between subsequent passages might be expected. However, the maximum roughness is observed at the intermediate passage value. Such an effect can be explained by taking into account that the porosity observed here is presumably a result of the cavitation process, and this phenomenon does not monotonously depend on the delivered energy [23]. The absolute roughness values in earlier works on laser-irradiated titanium and its alloy were in the range of 15–65 nm [22], 70–350 nm [16], 0.2–3.3 µm [37], 50–230 um [39], and 0.42–6.94 µm [36], close to those measured here. Despite that, the exceptionally high roughness appearing with the intermediate laser power cannot be explained at the present time. Both experiments and modeling are planned for the future to verify this effect and to propose the processes and mechanisms resulting in such behavior. Among them, melting and remelting, together with the associated phase transformations, the occurrence and dissolution of intermetallic phases, the temperature- and concentration-dependent diffusion of elements in liquid, and the appearance of stresses, among others, will be considered.

### 3.5. Nanoindentation Test

Table 6 presents the results of hardness, reduced Young’s modulus, and the maximum depth, which are average values from the measurements obtained in the nanoindentation test. For each material, a load–strain double loop was plotted from the nanoindentation test (Figure 4). The slope angle of the unloading curve determines the plastic–elastic behavior of the tested material [40]. The hardness H and reduced Young’s modulus E_r_ were determined, and Young’s modulus was calculated. Based on that, the ratios of H/E_r_ and H^3^/E_r_^2^ were calculated.

The use of laser processing has a large influence on the presented values. The samples subjected to laser remelting show higher hardness and Young’s modulus than the base material of the Ti13Nb13Zr alloy. Both values increase as the laser beam spacing increases. Compared to the other results, in which the hardness ranges between 220 and 770 HV [18,22,37] or 8 GPa [25,41], these values are comparable. As regards Young’s modulus, the present value is higher than previously reported after laser treatment, which was 40 to 60 GPa at 3 µm [23].

To ensure adequate tribological resistance, the values of the ratios should be as high as possible. The values of the parameters H/E_r_ and H^3^/E_r_^2^ for the treated samples should be greater than for the base alloy material [42].

Based on the results in Table 7, it is noted that the values of the ratios for laser-remelted samples are higher than for the base material. In the case of the wear resistance factor, the differences are small. A significant increase in the values of the ratios determining the material’s resistance to plastic deformation is observed. The most wear-resistant is the LT 30 sample with a scan line spacing of 30 µm. It should be noted, however, that its hardness parameter could be subject to a measurement error. In this case, the best resistance is shown by the sample with the largest laser beam spacing, and the values of both coefficients will be characterized by an upward trend. Based on the determined ratios after nanoindentation, it can be concluded that the applied treatment increased the tribological resistance of the Ti13Nb13Zr titanium alloy.

The improvement of the wear resistance of titanium and its alloys is usually achieved by the laser-associated surface modification of substrates, with a decrease in wear mass loss of over 90%. Such features were demonstrated by the laser cladding of WC-Co composites [43], the use of the LENS technique to create silica coatings on titanium [44], laser alloying resulting in Ti–C and Ti–Si coatings on a Ti–6Al–4V substrate [45], and the laser nitrogen alloying of Ti–35.3Nb–7.3Zr–5.7Ta alloy, as measured by the pin-on-plate sliding test [46]. Here, similar effects were observed after only direct laser writing, i.e., without any supplementary atoms or ions. Thus, even the proper topography and morphology can successfully improve the wear resistance, even if the effect is much smaller than after the laser-enhanced cladding of coatings or laser alloying. It might be questioned whether the nanoindentation test can be considered a reliable tool for assessing wear resistance as compared to the commonly used pin-on-disc technique or the use of a tribometer. There is a lack of such comparisons in the literature, but an assumption about the importance of this technique was already postulated in [47], claiming that higher H/Er, H^3^/Er^2^, and also We/Wt (elastic recovery index) values prove better wear and impact resistance. However, it is noted that even though the initial surface state (roughness, porosity, presence of oxide layer) might have a significant impact on the wear resistance of the alloy, the applied laser treatment dramatically transforms the surface layer because of a variety of described processes, and the previous treatment has almost no influence on its characteristics. In addition, both parameters characterize resistance to elastic and plastic deformation, but they are only very approximate measures of wear resistance, and these properties should be determined by other specific techniques, such as pin-on-disc or linear tribometry methods. The nanoindentation test can then be applied only to track the tendency to wear and to compare the effects of different surface treatments.

## 4. Conclusions

Laser remelting with the direct writing method had a significant impact on the change in the properties of the Ti13Nb13Zr alloy. Microstructural changes with a different topography and significant surface porosity were observed in each of the samples. The laser scan lines were visible. However, after the analysis of the EDS spectra, no effect of the laser beam spacing on the chemical composition was observed.

The thickness of the obtained surface layers decreased with the increasing spacing of the laser beam. A zonal structure of the surface layer of the titanium alloy after laser treatment was found. The significant fragmentation of the coniferous structure in the near-surface zone was observed. The thicknesses of all zones decreased with increasing laser beam spacing.

The microhardness of the near-surface zone measured by the Vickers method increased with increasing beam spacing. This means that there is a relationship between the width of the beam spacing and the thickness and hardness of the layers: the wider the beam, the smaller the thickness of the surface layer, and at the same time, the greater the hardness of the laser-processed material.

The nanohardness, Young’s modulus, and maximum cavities were interrelated. The first two parameters were characterized by an upward trend, while the values of the maximum depth decreased with increasing hardness.

An increase in the coefficients was observed for wear resistance, determined by the material’s resistance to plastic deformation in the nanoindentation test. Higher values of these parameters showed the increased tribological resistance of the Ti13Nb13Zr alloy.

The applied laser treatment with the direct writing method had a positive effect on the tribological resistance of the Ti13Nb13Zr titanium alloy.

## Figures and Tables

**Figure 1 materials-16-04834-f001:**
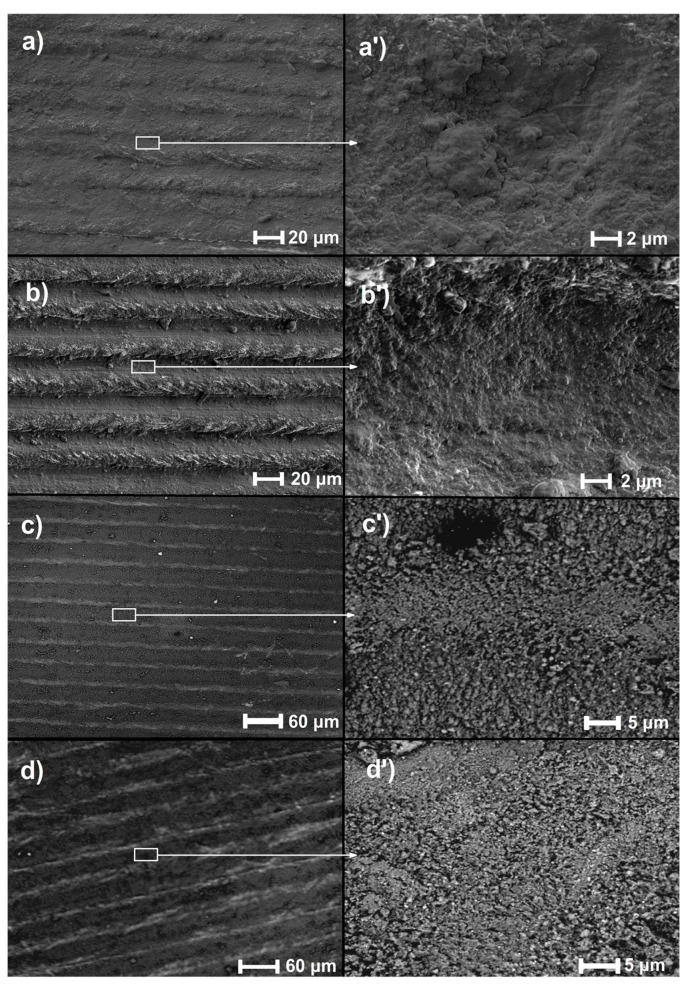
Topographies of laser-remelted samples: (**a**,**a’**) LT 20, (**b**,**b’**) LT 30, (**c**,**c’**) LT 40, (**d**,**d’**) LT 50.

**Figure 2 materials-16-04834-f002:**
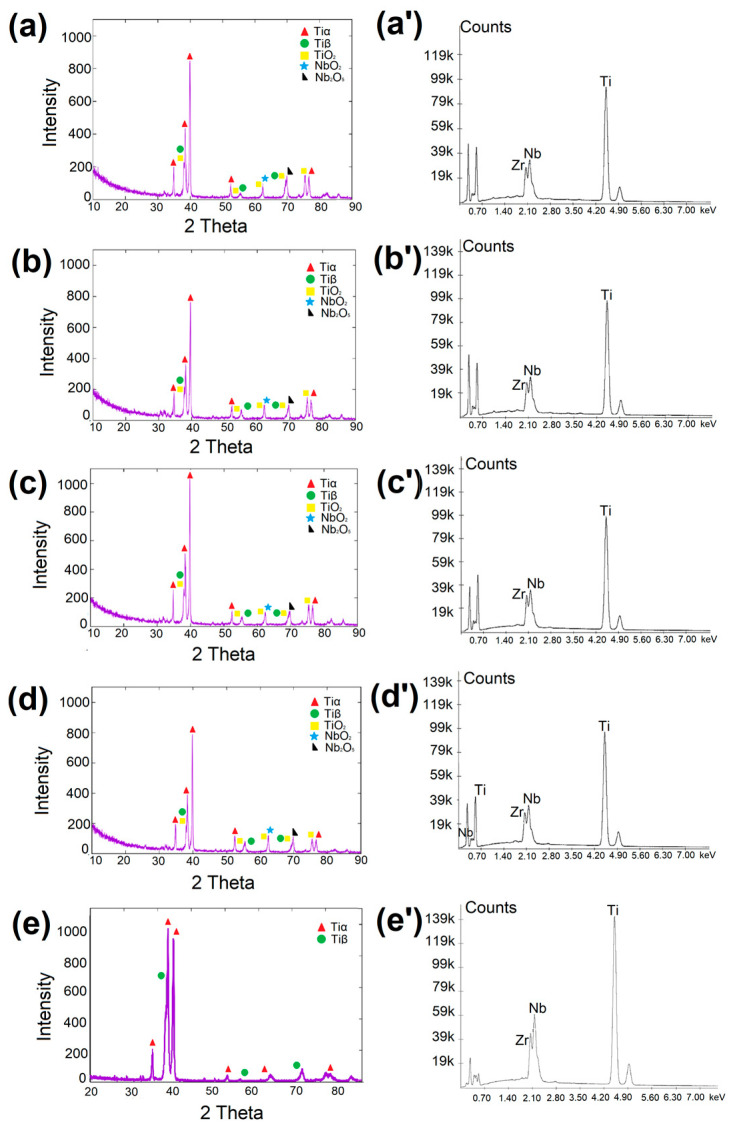
Phase and chemical composition of samples: (**a**) LT 20, (**b**) LT 30, (**c**) LT 40, (**d**) LT 50, (**e**) BM.XRD. (**a’**) LT 20, (**b’**) LT 30, (**c’**) LT 40, (**d’**) LT 50, (**e’**) BM.EDS.

**Figure 3 materials-16-04834-f003:**
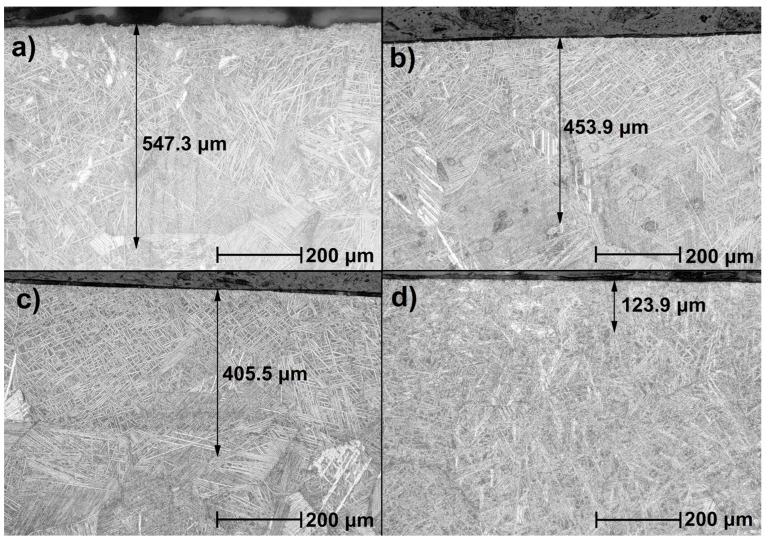
Microstructure of remelted samples: (**a**) LT 20, (**b**) LT 30, (**c**) LT 40, (**d**) LT 50. The black arrows indicate the melted layers’ thicknesses.

**Figure 4 materials-16-04834-f004:**
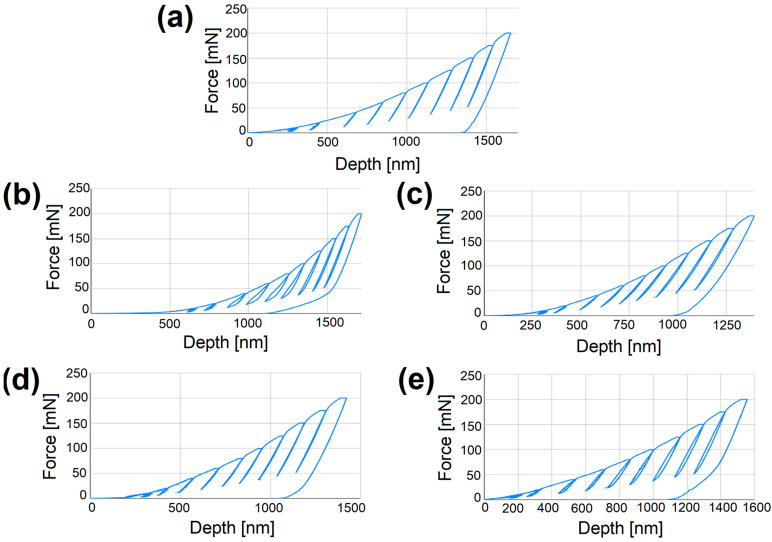
Indentation curves: (**a**) BM, (**b**) LT 20, (**c**) LT 30, (**d**) LT 40, (**e**) LT 50.

**Table 1 materials-16-04834-t001:** Chemical composition of Ti13Nb13Zr (based on manufacturer’s certificate).

C	Fe	N	O	Zr	Nb	H	S	Ti
% by weight								
0.035	0.085	0.019	0.078	13.49	13.18	0.055	<0.001	73.06

**Table 2 materials-16-04834-t002:** Chemical composition of Ti13Nb13Zr as base material and after laser modification (based on EDS analysis) with standard deviations.

	BM	LT 20	LT 30	LT 40	LT 50
% by weight					
Zr	11.03 ± 0.11	14.73 ± 0.12	14.26 ± 0.09	14.93 ± 0.27	14.61 ± 0.28
Nb	14.13 ± 0.06	14.67 ± 0.10	13.55 ± 0.12	13.93 ± 0.28	16.43 ± 0.28
Ti	74.84 ± 0.08	70.60 ± 0.09	72.18 ± 0.08	72.14 ± 0.11	70.96 ± 0.11

**Table 3 materials-16-04834-t003:** Layer thicknesses with standard deviations.

Sample	Thickness ± SD μm
LT 20	547.3 ± 6.8
LT 30	453.9 ± 6.2
LT 40	405.5 ± 11.4
LT 50	123.9 ± 2.8

**Table 4 materials-16-04834-t004:** Microhardness of subsurface layers with standard deviations.

Sample	Microhardness ± SD HV 0.01 (GPa)
BM	2.489 ± 0.13
LT 20	3.034 ± 0.28
LT 30	3.632 ± 0.26
LT 40	3.720 ± 0.40
LT 50	3.867 ± 0.10

**Table 5 materials-16-04834-t005:** The roughness of remelted samples with standard deviations.

Sample	Roughness + SD R_a_ (µm)
BM	0.02 ± 0.01
LT 20	0.69 ± 0.02
LT 30	1.74 ± 0.03
LT 40	0.44 ± 0.01
LT 50	0.24 ± 0.02

**Table 6 materials-16-04834-t006:** Hardness and Young’s moduli of surface layers with standard deviations.

Sample	Hardness (GPa)	Maximum Indent Depth ± SD (nm)	Reduced Young’s Modulus ± SD (GPa)	Young’s Modulus E ± SD (GPa)
BM	3.45 ± 0.19	1070.9 ± 453.1	107.6 ± 3.5	103.4 ± 1.7
LT 20	4.18 ± 1.29	989.5 ± 410.7	116.1 ± 24.9	113.1 ± 26.1
LT 30	5.26 ± 4.18	949.4 ± 394.1	118.3 ± 25.8	115.2 ± 21.0
LT 40	4.9 ± 1.52	892.4 ± 340.5	139.8 ± 15.7	138.7 ± 14.1
LT 50	5.44 ± 2.12	883.9 ± 381.2	146.1 ± 26.3	146.2 ± 27.1

**Table 7 materials-16-04834-t007:** Mean values of the ratios H/E_r_ and H^3^/E_r_^2^ with standard deviations.

Sample	H/E_r_ ± SD (-)	H^3^/E_r_^2^ ± SD (MPa)
BM	(3.21 ± 0.1) × 10^−2^	3.57± 0.41
LT 20	(3.6 ± 0.49) × 10^−2^	5.42 ± 2.52
LT 30	(4.44 ± 1.59) × 10^−2^	10.38 ± 16.43
LT 40	(3.51 ± 0.53) × 10^−2^	6.02 ± 3.54
LT 50	(3.72 ± 0.95) × 10^−2^	7.54 ± 7.25

## Data Availability

Not applicable.

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
