# Peer review of "Mechanical Properties and Wear Susceptibility Determined by Nanoindentation Technique of Ti13Nb13Zr Titanium Alloy after “Direct Laser Writing”"

_materials, 2023, doi:10.3390/ma16134834_

Round 1

Reviewer 1 Report

Dear authors,

Overall, good research and a good manuscript that deserves to be published. Nevertheless, there are some minor issues that should be addressed prior to publication:

1) The title “Mechanical properties and wear resistance of…” In fact, only properties that indicate improvement of wear resistance were determined, but no wear tests were done to provide the final proof. Therefore, I suggest crossing out “wear resistance”.

2) The abstract should not only describe what was done but also contain the most important results.

3) Line 99: Polishing with a diamond paste – which granulation?

4) Line 107: “…from 20 mm to 50 mm.   Millimeters? Probably micrometers.

5) Line 139: “Próbki poddane zostaÅ‚y obciążeniu 10 gf przez 10 sekund.”

6) Line 140: “…samples were subjected to a load of 10 gf for 10 seconds.” Instead of “gf”, perhaps HV 0.01 would be better. HV 0.01 basically tells the same but is the appropriate standard designation.

7) Lines 151-152: “…in the literature…”. Please quote the source. Modulus calculated according to equation (1) is in fact the indentation modulus, which, according to ISO 14577-1, can be comparable to Young’s modulus, but not always is. Quote: “Significant differences between the indentation modulus and Young's modulus can occur, if either pile-up or sink-in is present.”  Can in case of Ti13Nb13Zr alloy we be sure that indentation modulus corresponds to Young’s modulus?

8) Figure 1: The micrographs are too dark. Editing with photo editing software will help.

9) Table 2: How can changes of chemical composition due to laser treatment be explained? Zr content increased for over 3 wt.% (approx. 30 % of original content). Please provide a short explanation/discussion.    

10) Line 291-292: “The applied treatment increased the tribological resistance of the Ti13Nb13Zr titanium alloy.” I suggest to cross-out this sentence, because this is not a fact until it is proven by tribological tests, although everything points to it – as you correctly stated in the lines 302-307.

Author Response

First, we would like to warmly thank the reviewer for his/her helpful remarks which certainly have contributed to the appearance of the improved manuscript. These remarks and our answers are shown below. Besides, all changes were made in the manuscript in the Follow changes style.  We hope that they will be sufficient to recommend our paper for publication.

  1. The title “Mechanical properties and wear resistance of…” In fact, only properties that indicate improvement of wear resistance have been determined, but no wear tests were done to provide the final proof. Therefore, I suggest crossing out “wear resistance”.

Answer: After your suggestion, we propose a title: “Mechanical properties and wear susceptibility determined by nanoindentation technique for the Ti13Nb13Zr titanium alloy after "direct laser writing" treatment.

  1. The abstract should not only describe what was done but also contain the most important results.

Answer: The most important research results have been added in the abstract – Lines 28-31.

  1. Line 99: Polishing with a diamond paste – which granulation?

Answer: The samples were polished with a diamond paste with a granulation of 3 µm. This has been added in the text – Line 133.

  1. Line 107: “…from 20 mm to 50 mm.”   Millimeters? Probably micrometers.

Answer: It has been changed to µm - Line 146.

  1. Line 139: “Próbki poddane zostaÅ‚y obciążeniu 10 gf przez 10 sekund.”

Answer: The sentence has been deleted.

  1. Line 140: “…samples were subjected to a load of 10 gf for 10 seconds.”Instead of “gf”, perhaps HV 0.01 would be better. HV 0.01 basically tells the same but is the appropriate standard designation.

Answer: It has been changed as suggested – Line 179.

  1. Lines 151-152: “…in the literature…”. Please quote the source.

Answer: The reference for values of Poisson`s coefficient for the investigated alloy is by [31] as cited in the text (line 197).

  1. Modulus calculated according to equation (1) is in fact the indentation modulus, which, according to ISO 14577-1, can be comparable to Young’s modulus, but not always is. Quote: “Significant differences between the indentation modulus and Young's modulus can occur, if either pile-up or sink-in is present.”  Can in case of Ti13Nb13Zr alloy we be sure that indentation modulus corresponds to Young’s modulus?  example in Figure 3.

Answer: Both values for diamond, which the indenter is made of, are usually cited in wide limits, depending on crystallographic features, as E 1050-1200 GPa and ν 0.10-0.29 [https://www.matweb.com/search/DataSheet.aspx?MatGUID=d8d230a8d9664bc390199dab7bc56e1e&ckck=1]. Besides, in the text, the information that both values have been given by the manufacturer  has been added (previous lines 196 and 180). We agree that the (reduced) indentation modulus, calculated (surface) Young`s modulus and bulk Young`s modulus can have slight or significantly different values. Therefore, the researchers using nanoindentation tests usually only compare the effects of some determinants on elastic modulus values as we have also done, and we have never said that the calculated values might be taken as bulk ones.

  1. Table 2: How can changes of chemical composition due to laser treatment be explained? Zr content increased for over 3 wt.% (approx. 30 % of original content). Please provide a short explanation/discussion.    example in Figure 3.

Answer: We have included the following explanation in the text, beginning from the present line 227:

“The relative increase in contents of Zr and Nb at the expense of Ti can be easily explained as titanium has the highest vaporization pressure among three alloying elements, and its loss during laser treatment is certainly the highest”.

Please advise us whether we should also add a plausible reference, e.g.,: Zhang, G.; Chen, J.; Zheng, M.; Yan, Z.; Lu, X.; Lin, X.; Huang, W. Element Vaporization of Ti-6Al-4V Alloy during Selective Laser Melting. Metals 2020, 10(4), 435. https://doi.org/10.3390/met10040435.9.

  1. Line 291-292: “The applied treatment increased the tribological resistance of the Ti13Nb13Zr titanium alloy.”I suggest to cross-out this sentence, because this is not a fact until it is proven by tribological tests, although everything points to it – as you correctly stated in the lines 302-307.

Answer: The sentence was reworded to: ”Based on the determined ratios after nanoindentation, it can be concluded that the applied treatment increased the tribological resistance of the Ti13Nb13Zr titanium alloy” – Lines 362-364.

Reviewer 2 Report

After reviewing the manuscript, I to my regret have to not recommend it for publication. First, I have many questions about the methodology for conducting the experiment. Also, there are no discussions of the results of the experiments or they are very superficial, and the research task and the result of the research are not clearly formulated. Below are the main notes.

1. The introduction describes in detail the various methods of laser surface treatment, but there is no information about the direct laser writing method studied in the article. What is the opportunity of using this method in comparison with the already studied methods of laser surface treatment? What advantages of its use are? What effect on the surface do you expect to see?

2. There is no justification why the Ti13Nb13Zr alloy was chosen as the studied material. What did the alloy rod represent? What is the technology of its production? Was any heat treatment carried out further? Hot-rolled rod in its initial state and after heat treatment can have a large difference in hardness. In addition, the Ti13Nb13Zr alloy can have superelasticity depending on its phase state.

3. There are numerous errors in the text – some paragraphs are written in various font sizes and have different line spacing. In line 139, the phrase is not in English.

4. Figure 1 – Please, indicate enlarged fragments. Are these lines the grooves or between grooves?

5. Table 2 – How should BM be identified? Base material? The text contains only native material.

6. Is the test sample a part of the rod (a cylindrical sample in the form of a tablet)? How then does the cross section of the sample look like? Or maybe it is a longitudinal section? It can demonstrate the thickness of the treated layer.

7. How was the surface prepared for nanoindentation? Was the study carried out using a metallographic section?

8. Why was the chemical composition of the treated samples studied on the surface, but not in the surface layer of metallographic sections?

9. After laser treatment, the content of zirconium increased on the surface of the samples. Why then are there no peaks of zirconium or zirconium oxide on the X-ray diffraction patterns? It is also necessary to add X-ray patterns of the initial material before laser treatment for comparison.

10. How exactly did you determine the thickness of the laser-cut layer? In figure 3 a and b there are large grains in the area highlighted by the arrow. One part of the grains is partially located in the area of the arrow, and another one (especially in figure b) is in the base of the alloy. In figure d, the structure under the arrow has no difference from the structure in the area of the arrow.

11. The explanation for the huge difference in roughness of sample LT 30 is not very clear. What porosity are we talking about? How was it measured? Isn't there porosity in other samples? Line 312 indicates significant porosity in all samples. Pores are visible on the structure of the surface layer, for example in Figure 3.

12. There is no clear understanding in the manuscript what exactly the authors want to achieve by using direct laser writing: melting of the surface layer and rapid crystallization with the formation of a fine-dispersed structure or a required roughness in the form of grooves/steps? Or all of that? The authors use nanoindentation as a way to determine the wear resistance of the alloy after direct laser writing and recognize this approach as not quite standard, but the application technique itself raises questions. According to the text, nanoindentation was carried out on polished sections in order to study the hardness of the surface layer with a fine-dispersed acicular microstructure and the base of the alloy. Based on these measurements, a conclusion is made about the increased wear resistance of the treated alloy. But the alloy surface itself contains roughness, significant porosity and various oxides. All this will have a significant impact on the wear resistance of the alloy. The authors do not take these factors into account.

Author Response

First, we would like to warmly thank the reviewer for his/her helpful remarks which certainly have contributed to the appearance of the improved manuscript. These remarks and our answers are shown below. Besides, all changes were made in the manuscript in the Follow changes style. We hope that they will be sufficient to recommend our paper for publication.

  1. The introduction describes in detail the various methods of laser surface treatment, but there is no information about the direct laser writing method studied in the article. What is the opportunity of using this method in comparison with the already studied methods of laser surface treatment? What advantages of its use are? What effect on the surface do you expect to see?

Answer: The introduction has been modified and information about the used method has been added (Lines 49-80). The authors, using this method of laser processing, assumed that it would increase the hardness of the material, and thus resistance to abrasive wear. In addition, changes in surface roughness were expected, which in further studies could have a positive effect on the adhesion of hydroxyapatite coatings.

  1. There is no justification why the Ti13Nb13Zr alloy was chosen as the studied material. What did the alloy rod represent? What is the technology of its production? Was any heat treatment carried out further? Hot-rolled rod in its initial state and after heat treatment can have a large difference in hardness. In addition, the Ti13Nb13Zr alloy can have superelasticity depending on its phase state.

Answer: The Ti13Nb13Zr alloy was chosen for the study due to its low Young's modulus and much more favorable chemical composition compared to the commonly used Ti6Al4V alloy in the context of medical applications. The rod was delivered in an annealed state, and this information was included in the article (Line 135). Recognizing the potential differences in material hardness between the initial state and after laser modification, it was measured in both cases, and the results were included in Table 6."

  1. There are numerous errors in the text – some paragraphs are written in various font sizes and have different line spacing. In line 139, the phrase is not in English.

Answer: The text has been corrected according to MDPI Styles. The phrase in line 139 was deleted.

  1. Figure 1 – Please, indicate enlarged fragments. Are these lines the grooves or between grooves?

Answer: Enlarged areas are indicated in Figure 1. These lines are grooves (brighter lines).

  1. Table 2 – How should BM be identified? Base material? The text contains only native material.

Answer: The description of the base material BM has been unified.

  1. Is the test sample a part of the rod (a cylindrical sample in the form of a tablet)? How then does the cross section of the sample look like? Or maybe it is a longitudinal section? It can demonstrate the thickness of the treated layer.

Answer: The test sample was cut from a rod, it was a cylindrical sample in the form of a tablet. The assessment of the thickness of the melted surface layers was made on cross-sections to the laser beam.

  1. How was the surface prepared for nanoindentation? Was the study carried out using a metallographic section?

Answer: Nanoindentation tests were performed from the surface of the samples. Samples after laser treatment were directly tested, without any additional preparation. On the other hand, a sample of the native material was previously prepared as before laser processing ‘The treatment was performed using sandpaper with a decreasing gradation of grain from 180 to 2000. The final stage was polishing with a diamond paste’.

  1. Why was the chemical composition of the treated samples studied on the surface, but not in the surface layer of metallographic sections?

Answer: The authors performed an analysis of the chemical composition on the melted surfaces and at this stage of the research the chemical composition on the cross-sections was not taken into account, we will check it in the future.

  1. After laser treatment, the content of zirconium increased on the surface of the samples. Why then are there no peaks of zirconium or zirconium oxide on the X-ray diffraction patterns? It is also necessary to add X-ray patterns of the initial material before laser treatment for comparison.

Answer: In Figure 2 added X-ray patterns of the base material for comparison. The study of the phase composition showed the presence of α and β phases. The lack of peaks from zirconium and zirconium oxide can be explained by the fact that zirconium is a natural stabilizer of the α and β phases and occurs in solid solutions.

  1. How exactly did you determine the thickness of the laser-cut layer? In figure 3 a and b there are large grains in the area highlighted by the arrow. One part of the grains is partially located in the area of the arrow, and another one (especially in figure b) is in the base of the alloy. In figure d, the structure under the arrow has no difference from the structure in the area of the arrow.

Answer: The surface layers obtained after the DLW show a zonal structure, they consist of the near-surface, middle and transition zones. The individual zones are not even. In the first zone, the smallest needles are observed, it is also the thinnest, in the middle their size increases, while in the transitional zone, there are larger ones coming from the substrate material and finer ones from the middle zone. Therefore, larger needles are also visible in these areas. Figure 3 shows layer thickness values averaged from six measurements. Figure 3 has been changed to more clearly show the structural changes on the cross-section in Fig 3d.

  1. The explanation for the huge difference in roughness of sample LT 30 is not very clear. What porosity are we talking about? How was it measured? Isn't there porosity in other samples? Line 312 indicates significant porosity in all samples. Pores are visible on the structure of the surface layer, for example in Figure 3.

Answer: We have carefully checked the research protocols, but there is no mistake, the exceptionally high roughness appears at the intermediate laser power. We have not measured porosity, we only see the pores. However, we have no idea yet, how to explain this effect which needs further both experiments and modeling in the future. Therefore, we would not like to say more at the moment, only such text has been inserted in the manuscript beginning from the present line 323:

“Despite that, the exceptionally high roughness appearing at the intermediate laser power cannot be explained at the present time. Both experiments and modeling are planned for the future to verify this effect and if confirmed to propose the processes and mechanisms resulting in such behavior. Among them, melting and remelting together with associated phase transformations, an occurrence and dissolution of intermetallic phases, temperature and concentration-dependent diffusion of elements in liquid, an appearance of stresses, and others will be considered”.

  1. There is no clear understanding in the manuscript what exactly the authors want to achieve by using direct laser writing: melting of the surface layer and rapid crystallization with the formation of a fine-dispersed structure or a required roughness in the form of grooves/steps? Or all of that?

Now, in the Introduction, we can only repeat why we have decided to apply the DLW technique and what we have expected as it has been quite a novel approach., Therefore, even based on state-of-the-art, we have been sure that we could expect the melting of the surface layer followed by its rapid crystallization, but not whether a fine-dispersed structure or a desired roughness would appear. Therefore, we propose only a slightly modified text finishing the introduction:

“There have been a variety of different laser-associated treatments focused on an increase in several mechanical properties, first of all, hardness and wear properties. This research is a novel approach in such a field as it characterizes the seldom applied, for titanium and other metals, technique of direct laser writing. This technique is promising as presumably not causing much damage to the surface layer. The aim of this research is to verify whether DLW can have a significant effect on the mechanical behavior of Ti alloy at a limited rebuilding of the surface layer.  The performed surface modification was made at different distances between the laser beam scans, the mechanical behavior was characterized and discussed based on changes in microstructures. The present investigation proves that this form of laser treatment can effectively and positively affect surface properties”.

  1. The authors use nanoindentation as a way to determine the wear resistance of the alloy after direct laser writing and recognize this approach as not quite standard, but the application technique itself raises questions. According to the text, nanoindentation was carried out on polished sections in order to study the hardness of the surface layer with a fine-dispersed acicular microstructure and the base of the alloy. Based on these measurements, a conclusion is made about the increased wear resistance of the treated alloy. But the alloy surface itself contains roughness, significant porosity and various oxides. All this will have a significant impact on the wear resistance of the alloy. The authors do not take these factors into account.

Answer: We have a little different opinion. The surface layer of an alloy before any treatment has its own specifics and we do not neglect it. It has a roughness resulting from applied mechanical treatment, small in this case; it has no significant porosity as it is made by foundry manufacturing followed by heat treatment; it has a single oxide layer which can be composed of anatase, anatase, or anatase and rutile. All of it is well-known, but any surface treatment, either mechanical, chemical, or physical, either rebuilding the surface layer by, e.g. any laser treatment, or deposition of ions, atoms, or coatings, changes properties of the surface, more or less depending on its previous history.  An applied laser treatment dramatically transforms the surface layer because of a variety of described processes, and the previous treatment does almost no influence its characteristics. However, taking into account your critiques, we have inserted in our manuscript the following text in present line 380.There are some suppositions that indexes H/R and H

“However, it is to note that even if the initial surface state (roughness, porosity, presence of oxide layer) might have a significant impact on the wear resistance of the alloy, applied laser treatment dramatically transforms the surface layer because of a variety of described processes, and the previous treatment does almost no influence its characteristics. Besides, both parameters characterize resistance to elastic and plastic deformation, but they are only very approximate measures of wear resistance, these properties should be determined by other specific techniques such as, e.g., pin-on-disc or linear tribometry methods. The nanoindentation test can be then applied only to track the tendency to wear and to compare the effects of different surface treatments”. 

Reviewer 3 Report

Referee report on Ms. ID: materials-2456448

Title:  Mechanical properties and wear resistance of surface layer formed on titanium alloy Ti13Nb13Zr by “direct laser writing" treatment

by Magdalena Jażdżewska, et al.

Ti-based alloys find a wide range of applications starting from aerospace and military industries, mobile phones, to medical prostheses and implants, etc. However, the use of such alloys in medicine encounters difficulties related to low biological activity, the possibility of metallosis and related inflammations or other allergic reactions. In order to avoid these disadvantages or at least to improve the required properties of the alloys, not only new materials are being researched, but scientists are also working on a way to modify the surface layers of existing alloys used in medicine. In this regard, the topic of this article is novel and the presented results are of scientific and practical interest. The authors propose a new approach to surface modification of Ti13Nb13Zr alloys by applying direct laser writing to improve the mechanical properties of the surface layers. The change in the structure and properties with the laser exposure are traced by SEM, EDAX, XRD and nanoindentation measurements. The article is easy to read and the experiments are well organized and clearly described. There are a few remarks which should be considered by the authors.

My opinion is that the article can be published after a minor revision.

Here are some remarks and recommendation, as follows:

-          Concerning the EDAX data, the sum of the chemical components should be 100 weight%.  However, for LT30 it is <100 % (99.99%); for LT40 - > 100 % (101%) and for LT50 the sum is >100 % (102%). This might come from EDAX measurement errors, but it should be discussed. It should be indicated with what accuracy the weight fraction is determined and the errors should also be given in Table 2 accordingly.

-          What is the meaning of BM? If it is the native material, why is it abbreviated as BM? The meaning of the abbreviation BM should be given.

-          Concerning Fig. 2, the observed oxidation of Ti and Nb atoms might be caused by the DLW process but could be also already in the surface layer of the untreated material. The authors should clarify this in the article and, in support of their statement "Therefore, it seems that DLW processing is very subtle and does not cause much damage to the surface layer" the spectrum of the initial material should also be given.

-          Section 3.3 needs a better explanation of the results. The sentence in lines 243-245 “Also, an increase in hardness with a local decrease in energy flow can be described as a relative shift in depth of the hardest zone closer to the surface. ” is rather speculative and unclear. It cannot explain the observed increasing hardness and decreasing thickness with increase of scan spacing.

-          It is desirable that similar physical quantities are in the same dimension. I recommend converting the Vickers hardness number to SI units, i.e. to give the microhardness of the subsurface layers in Table 4 also in GPa.

-          The list of References should be checked. There are references with incomplete data, for example Ref. 32 - Materials, (2019) should be Materials, 2019, 12, 2964. See further refs 7,29,32,33,41.

The English needs some improvement, for example:

-          line 139; “Prуbki poddane zostaÅ‚y obciążeniu 10 gf przez 10 sekund. “ . Please, delete the sentence in Polish.

-          line 183, Table 2 shows the chemical composition of the samples...”;

-          line 190, “Figure 2 presents the phase and chemical composition of the samples..”;

-          line 207; “Figures 3 show the...” should read “Figure 3 shows the.....” ; etc.

The English needs some improvement, for example:

-          line 139; “Prуbki poddane zostaÅ‚y obciążeniu 10 gf przez 10 sekund. “ . Please, delete the sentence in Polish.

-          line 183, Table 2 shows the chemical composition of the samples...”;

-          line 190, “Figure 2 presents the phase and chemical composition of the samples..”;

-          line 207; “Figures 3 show the...” should read “Figure 3 shows the.....” ; etc.

Author Response

First, we would like to warmly thank the reviewer for his/her helpful remarks which certainly have contributed to the appearance of the improved manuscript. These remarks and our answers are shown below. Besides, all changes were made in the manuscript in the Follow changes style.  We hope that they will be sufficient to recommend our paper for publication.

  1. Concerning the EDAX data, the sum of the chemical components should be 100 weight%.  However, for LT30 it is <100 % (99.99%); for LT40 - > 100 % (101%) and for LT50 the sum is >100 % (102%). This might come from EDAX measurement errors, but it should be discussed. It should be indicated with what accuracy the weight fraction is determined and the errors should also be given in Table 2 accordingly.

Answer: The authors added errors in Table 2. EDAX analysis is actually a qualitative study about the presence or absence of any element, it was used for comparison, however, the results obtained are subject to errors resulting from, among others, overlapping peaks or the inability to analyze elements with atomic number ≥5.

  1. What is the meaning of BM? If it is the native material, why is it abbreviated as BM? The meaning of the abbreviation BM should be given.

Answer: Base material was abbreviated as BM, by mistake of the authors, native material was used in the text, and it was corrected in the article. The meaning of BM has been corrected on lines 141 and 184.

  1. Concerning Fig. 2, the observed oxidation of Ti and Nb atoms might be caused by the DLW process but could be also already in the surface layer of the untreated material. The authors should clarify this in the article and, in support of their statement "Therefore, it seems that DLW processing is very subtle and does not cause much damage to the surface layer" the spectrum of the initial material should also be given.

Answer: In Figure 2(e), X-ray patterns of the base material have been added for comparison. The study of the phase composition showed the presence of α and β phases, there are no Ti or Nb oxide peaks.

  1. Section 3.3 needs a better explanation of the results. The sentence in lines 243-245 “Also, an increase in hardness with a local decrease in energy flow can be described as a relative shift in depth of the hardest zone closer to the surface. ” is rather speculative and unclear. It cannot explain the observed increasing hardness and decreasing thickness with increase of scan spacing.

Answer: Regretfully, accepting this remark we can now only change the section 3.3. to the following one:

“The results of microhardness measurements on cross-sections in the near-surface zone of the remelted layer are presented in Table 4. The microhardness of the Ti13Nb13Zr BM titanium alloy on the Vickers scale was 254 HV. As a result of laser treatment, the samples were hardened. The average hardness of the near-surface zone of the LT 20 sample was 3.034 GPa, LT 30 – 0.3632 GPa, LT 40 – 0.372 GPa, and LT 50 – 0.3867 GPa. The difference in hardness of the near-surface zone between the LT 30 and LT 40 samples is insignificant. Based on the results presented in Table 4, it can be concluded that the hardness of the near-surface zone increases with the increase of the laser beam spacing. The lower energy flow makes the zone-affected zone depth smaller, and the higher energy makes it greater. Therefore, the highest hardness appears at some distance from the surface, e.g., Katahira et al. [28] observed a surface hardness of 1.618 GPa, which increased to  2.511 GPa at 3 μm and further decreased to 9 μm. It is then assumed that the hardness peak is visible at some depth, likely increasing with lowering energy. However, it speculative assumption, and the effects of possible dissolution and precipitation of intermetallic phases related to the power and temperature, or of thermal stresses should be taken into account. This unsolved now observation will be studied in the future”.        

  1. It is desirable that similar physical quantities are in the same dimension. I recommend converting the Vickers hardness number to SI units, i.e. to give the microhardness of the subsurface layers in Table 4 also in GPa.

Answer: The units in Table 4 have been converted and reported in GPa.

  1. The list of References should be checked. There are references with incomplete data, for example  32- Materials, (2019) should be Materials201912, 2964. See further refs 7,29,32,33,41.

Answer: References have been checked and listed corrected.

  1. The English needs some improvement, for example:

 line 139; “Prуbki poddane zostaÅ‚y obciążeniu 10 gf przez 10 sekund. “ . Please, delete the sentence in     Polish.

 line 183, “Table 2 shows the chemical composition of the samples...”;

 line 190, “Figure 2 presents the phase and chemical composition of the samples..”;

 line 207; “Figures 3 show the...” should read “Figure 3 shows the.....” ; etc.

Answer: The manuscript has been thoroughly checked by the Grammarly software and all indicated errors have been deleted.

Round 2

Reviewer 2 Report

The authors significantly revised the manuscript, taking into account the comments, made the introduction and experimental methodology more understandable. Therefore, the article can be accepted for publication in its present form.